# Field theory for recurrent mobility

Mattia Mazzoli [1], Alex Molas[1], Aleix Bassolas [1], Maxime Lenormand [2], Pere Colet [1] & José J. Ramasco [1]

Understanding human mobility is crucial for applications such as forecasting epidemic spreading, planning transport infrastructure and urbanism in general. While, traditionally, mobility information has been collected via surveys, the pervasive adoption of mobile technologies has brought a wealth of (real time) data. The easy access to this information opens the door to study theoretical questions so far unexplored. In this work, we show for a series of worldwide cities that commuting daily flows can be mapped into a well behaved vector field, fulfilling the divergence theorem and which is, besides, irrotational. This property allows us to define a potential for the field that can become a major instrument to determine separate mobility basins and discern contiguous urban areas. We also show that empirical fluxes and potentials can be well reproduced and analytically characterized using the so-called gravity model, while other models based on intervening opportunities have serious difficulties.

[1] Instituto de Física Interdisciplinar y Sistemas Complejos IFISC (CSIC-UIB), Campus UIB, 07122 Palma de Mallorca, Spain. [2] Irstea, UMR TETIS, 500 rue JF Breton, 34093 Montpellier, France. Correspondence and requests for materials should be addressed to M.M. (email: mattia@ifisc.uib-csic.es) or to J.J.R. (email: jramasco@ifisc.uib-csic.es)

Human mobility has been studied for decades due to the relevant role it plays in a wide spectrum of applications including economic questions and living conditions[1–3], city structure[4,5], forecasting epidemic spreading[6–9], traffic demand and design of new infrastructure[10], or urban pollution and air quality[11]. Data on people migrations dates back at least to 1871 when the United Kingdom registered the difference in inhabitants during a decade[12]. More recently, in the last decades, census surveys in countries around the world have included a question on the tract of residence and that of work (see for instance the Supporting Information of[7] to find a list). Aggregating the home-work trips of the single individuals, one can define the so-called Origin-Destination (OD) matrices that for every pair $(i, j)$ collect the flow of people traveling from census tract $i$ to $j$, $T_{ij}$. These matrices are absolutely essential for transport planning since they encode trip demand. Census and specially dedicated surveys have dominated the area in terms of mobility data collection until a few years ago[13,14]. With the advent of the big data era, the availability of large-scale quick-updated data has notably increased. Passive sources such as mobile phone records or GPS-located messages in online social networks (Twitter, Foursquare, etc) have been employed to study mobility[15–20] and, in particular, to extract OD matrices (see also the recent reviews[14,21]). It is worth noticing that the quality of the OD matrices obtained from these new information and communication technologies (ICT) data sources have been confronted against the information provided by surveys with satisfactory results in urban areas at geographical scales larger than one square kilometer[18]. The wealth of new data opens the door to tackle and revisit relevant theoretical aspects concerning mobility flows that could not been boarded before.

From a theoretical perspective, two competing frameworks have been used for almost 80 years to characterize mobility flows: the gravity[22,23] and the intervening opportunity[24,25] models. Their main difference lies in the way in which the geographical distance affects the flows. While in the gravity model the flows decay with a certain deterrence function (usually, with an exponential or power law-like forms[26–29]), the intervening opportunity models depend on the "opportunities" or jobs enclosed within a given area. Since the opportunity distribution can be highly heterogeneous in space, the distance plays an indirect role on the final assignment of the trip destinations and, in turn, on the decay of the total flows[14,30]. A few years ago, it has been introduced the so-called radiation model as an evolution of the intervening opportunity concept in which the opportunity selected is supposed to be the best possible choice simplifying the statistic treatment, and the density of opportunities is related to the population[30,31]. This allows to write a closed formula for the probability of a trip to finish at a given geographical unit. Regarding the gravity model, its functional shape was proposed ad hoc, essentially inspired by Newton's law in which the populations act as masses[32,33], although it can be also recovered from maximal entropy arguments[34]. Moreover, the model can be developed further by taking into account the distinguishability of the trips[35–37]. Early after the gravity model introduction, the possibility of defining a potential was discussed[38] but the lack of reliable data prevented ulterior research in this direction.

Several works have focused on the comparison between the two families of models and their performance when compared with empirical data[39–47]. It is worth mentioning that a fair comparison requires to be carried out over the same type of mobility data (daily or sporadic trips behave differently) and with the same constraints. The constraints here refer to the amount of information provided to the model. The basic unconstrained models only include the population in the geographical units, while in the constrained versions the total in- or/and out-flows are also supplied[46].

In this work, we propose a method to define a mesoscopic vector field out of daily commuting data. This field turns out to be well-behaved, fulfilling Gauss's divergence theorem and being irrotational. Given that we are analyzing empirical information, these results are far from trivial and they reveal intrinsic features of aggregated daily human mobility. The existence of a well-behaved mesoscopic field is confirmed with both data from Twitter and census for large urban areas. By taking into account the irrotational nature of the field, we also define a potential for the mobility flows. This potential is a tool that will crucially contribute to controversial issues such as the functional definition of city limits[48] and the presence of polycenters[5]. After these first empirical results, we focus on which properties of the mesoscopic field can be reproduced by the models. In the case of the gravity, the fluxes over surfaces, rotational and potential empirical observations are well reproduced with an exponentially decaying deterrence function and they can be analytically obtained or approximated. The radiation model has, however, stronger difficulties to reproduce the empirical values of the fluxes.

## Results

**Definition of the vector field.** We obtain OD matrices between cells of $1 \times 1$ km$^2$ from Twitter and, where available, also from census data in several worldwide cities (see Supplementary Table 1 for a list of cities and Methods, below, for a description of the data cleaning procedure). We call $T_{ij}$ to the daily flow of commuters from cell $i$, home, to $j$, work. There can be flows between any pair of cells in the city. As defined, the OD matrix $T_{ij}$ contains only information on trips origin and final destination, not about trajectories or middle points visited. We then define a vector centered in $i$, $T_{ij} \overrightarrow{\mathbf{u}}_{ij}$, where $\overrightarrow{\mathbf{u}}_{ij}$ is the unit vector from $i$ to $j$. The vectors pointing to all destinations $j$ are then vectorially summed to obtain a resultant vector $\overrightarrow{\mathbf{T}}_i = \sum_j T_{ij} \overrightarrow{\mathbf{u}}_{ij}$ in every cell $i$ (see Fig. 1a). These vectors define a field in the space and they identify the mean outgoing mobility direction in every point. If the mobility is balanced in opposite directions, the vector $\overrightarrow{\mathbf{T}}_i$ can vanish. These equilibrium (Lagrange) points play an important role in the field theoretical framework. As an illustration, empirical fields for London and Paris are displayed in Fig. 1b, c, respectively. Further examples for other cities are shown in the Supplementary Figs. 27–42.

Drawing a parallel with classical field theories, $\overrightarrow{\mathbf{T}}_i$ can be divided by the "mass" of the origin cell $i$ (home-place) to define the vector field

$$\overrightarrow{\mathbf{W}}_i = \frac{\overrightarrow{\mathbf{T}}_i}{m_i} = \sum_{j \neq i} \frac{T_{ij}}{m_i} \overrightarrow{\mathbf{u}}_{ij}, \tag{1}$$

where the mass $m_i$ corresponds to the cell population considered in the analysis. The vector $\overrightarrow{\mathbf{W}}_i$, defined at the mesoscopic cell-size scale, is the main object of study in this work and it represents an average mobility per capita. Our data refers to commuters, either those calculated from Twitter or collected by the census. For practical reasons, we define the local mass $m_i$ as the total number of commuters residing in cell $i$. This means that $m_i = \sum_j T_{ij}$, with the sum including the term $j = i$. This definition allows us to apply a coherent treatment to all our databases and it is an approximation for the total workforce living in every cell. As shown in[46], the mass defined in this way yields better flow estimates than the actual cell population for both gravity and radiation models.

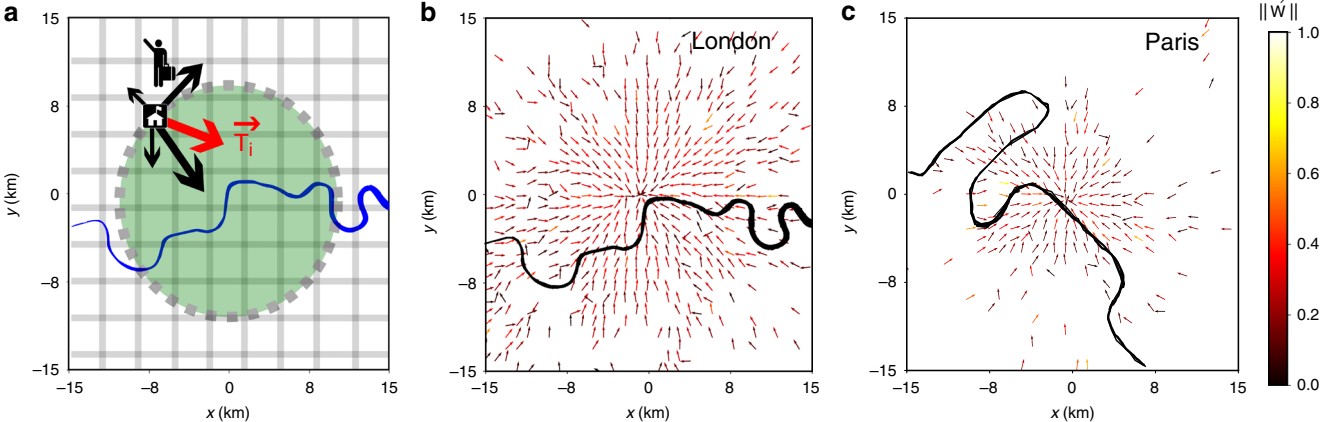

**Fig. 1** Empirical vector fields. **a** Sketch of the method to build the vector field. Each flow from cell $i$ to $j$ is a vector centered in $i$, with direction pointing to $j$ and whose modulus is equal to $T_{ij}$. Summing vectorially these vectors, we obtain $\overrightarrow{\mathbf{T}}_i$ and from it, dividing by the population in $i$, we get the vector field $\overrightarrow{\mathbf{W}}_i$. Commuters vector field W for the London (**b**) and Paris (**c**) areas. Colors represent the module of the field $||\overrightarrow{\mathbf{W}}_i||$ per cell

If instead of home to work, we consider the returning trip from work to home the picture does not change significantly. If the vectors $\overrightarrow{\mathbf{T}}_i$ are still defined at the residence cell, their sense reverses but the modulus remains unchanged. The spatial organization of the field is, therefore, invariant and it does not affect the results shown below (except for one sign). On the other hand, if instead of calculating the resultant vector at the residence place we define it at the working cell: $\overrightarrow{\mathbf{T}}'_j = \sum_i T_{ij}\overrightarrow{\mathbf{u}}_{ji}$ and $\overrightarrow{\mathbf{W}}'_j = \overrightarrow{\mathbf{T}}'_j / m_j$, the values of the vectors themselves modify at every location but the mesoscopic field behavior and the main properties studied below are robust (see Supplementary Note 13 and Supplementary Fig. 51).

**Empirical results**. Once the field is defined, we can calculate directly from empirical data the flux across any closed perimeter from the surface integral $\Phi_W^S = \oint d\ell\, \overrightarrow{\mathbf{n}}\, \overrightarrow{\mathbf{W}}$, where $\overrightarrow{\mathbf{n}}$ is the unit vector normal to the perimeter in each point and $d\ell$ the infinitesimal of length, and compare it with the volume integral of the divergence of $\overrightarrow{\mathbf{W}}$, $\Phi_W^V = \int dS\, \nabla\overrightarrow{\mathbf{W}}$, in the area enclosed inside the perimeter (where $dS$ is the infinitesimal of area). This allows us to assess whether the empirical vector field $\overrightarrow{\mathbf{W}}$ fulfils Gauss's Theorem of the Divergence or not. Gauss's theorem states that

$$\Phi_W^S = \oint d\ell\, \overrightarrow{\mathbf{n}}\, \overrightarrow{\mathbf{W}} = \int dS\, \nabla\overrightarrow{\mathbf{W}} = \Phi_W^V, \qquad (2)$$

and it implies that the field is generated by a source and that the fluxes through surfaces must respect conservation laws. The numerical estimations of the flux $\Phi_W$ as a function of the scale using both integrals are shown in Fig. 2 for London and Paris with two perimeter shapes: a circle and a square. As it can be seen, the agreement between both approaches is rather good with $R_P^2 = 0.96$ (circle) and $R_P^2 = 0.89$ (square) for London and $R_P^2 = 0.97$ (circle) and $R_P^2 = 0.80$ (square) for Paris. $R_P^2$ is obtained as the square of the Pearson correlation coefficient of both curves. We have run the same test in several cities with Twitter data. Supplementary Table 1 shows the list of coordinates of the central points of the perimeters in each city and Supplementary Table 2 the results of the comparisons. In most of the cases the values of $R_P^2$ are in the range 0.8–0.97 with only two exceptions that are, in any case, over 0.66. For completeness, the same operation has been performed with census data in London ($R_P^2 = 0.98$ both for the circle and the square) and in Paris ($R_P^2 = 1$ for the circle and

$R_P^2 = 0.98$ for the square) as can be seen in Supplementary Fig. 1. This implies that the field does indeed fulfil Gauss's theorem.

Similarly, we can compute the curl of the vector field directly out of the data (see Methods). The field $\overrightarrow{\mathbf{W}}$ is embedded in a $x$–$y$ plane and, therefore, $\nabla \times \overrightarrow{\mathbf{W}}$ has only a component on the z-direction. The outcome of $||\nabla \times \overrightarrow{\mathbf{W}}||$ using a colormap is depicted in Fig. 3a. The values of the curl modulus is of the order of $10^{-1}$ in $km^{-1}$. To evaluate whether this is small or large, we have defined a null model by randomly redirecting the angles of the vectors of each cell. The curl of the random model is of the same scale as the empirical field (Fig. 3b). For instance, calculating the dimensionless numbers $\int dS\, ||\nabla \times \overrightarrow{\mathbf{W}}||^2$ we obtain 21 for the empirical field and 45 for null model. Furthermore, the distribution of the original $\nabla \times \overrightarrow{\mathbf{W}}$ is similar to the random one, with a mixed between a delta distribution at zero and a symmetric exponential decay in the tails (Supplementary Fig. 43). This means that the values that we observe in the empirical curl are compatible with random fluctuations and the possibility of having a developed rotational structure in the field is rejected. The comparison with the modulus of the original field shows as well that the curl is 4 orders of magnitude smaller (Fig. 1b). All these evidences support the irrotational character of $\overrightarrow{\mathbf{W}}$ and allow us to define a potential for it. These results are further supported by the vectors $\overrightarrow{\mathbf{W}}$ angle analysis performed in Supplementary Note 11 (Supplementary Figs. 45–50).

Circular infrastructures are not so uncommon in cities, besides circular metro lines many highways are organized as concentric rings when there is no major geographical impediment as in Paris or London. One may, thus, wonder why typically we do not observe rotational components in the cities vector field. To have such components, it would be necessary to have an unbalanced flow of people living in an area and working in another over the ring following one of the rotation senses. At the scale that we are using, this is not seen anywhere in the cities under study. The main factor that could favor the emergence of rotational components is thus the segregation of land use. However, land use mixing is strong enough in large cities[49] to prevent this sort of loops in the mobility flows at mesoscopic scales, leading to hierarchical configurations of the mobility with a few clear attraction centers.

**Models**. There are two main modeling frameworks in the literature to characterize mobility flows: those based on

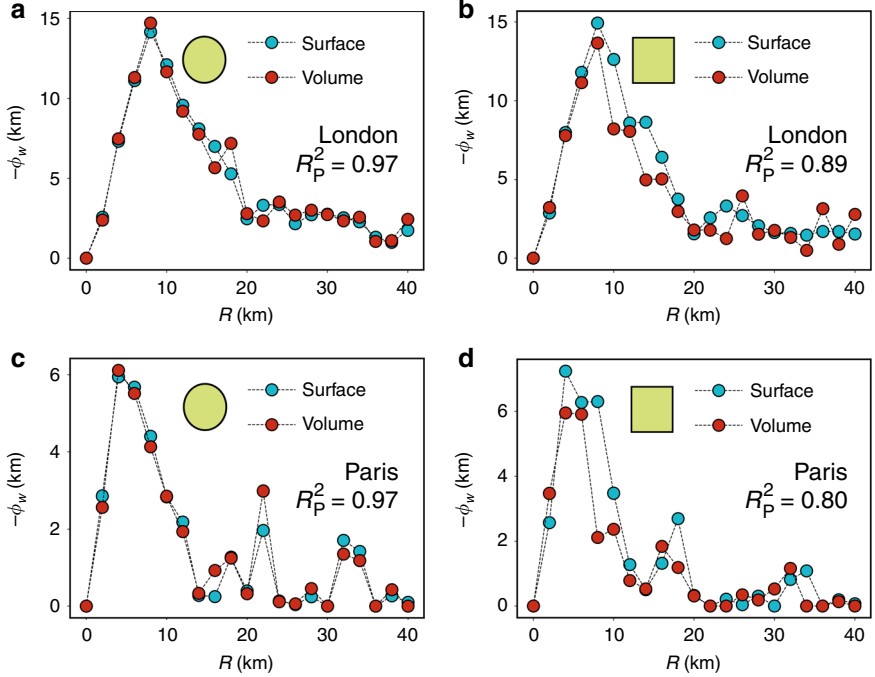

**Fig. 2** Gauss's theorem. In blue, the flux calculated as the surface integral of the vector field $\overrightarrow{\mathbf{W}}$ obtained from commuters going to work. In red, the volume integral of $\nabla \overrightarrow{\mathbf{W}}$. **a** Results in London for a circle of radius $R$ centered at Waterloo Bridge. **b** A square perimeter around London with the same center and side $2R$. In the figure, we show half the side as the $x$-axis variable to maintain the geographical scales similar to those of the circle radius. The same for Paris with a circle (**c**) and a square (**d**) centered at the Passage du Grand Cerf in the 2nd Arrondissement

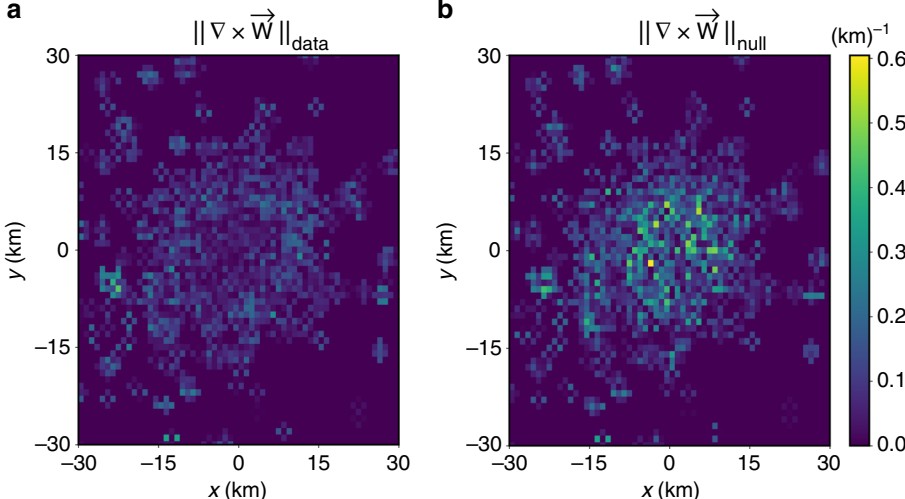

**Fig. 3** Curl of $\overrightarrow{\mathbf{W}}$. **a** The curl in London, the colors represent the module of $\nabla \times \overrightarrow{\mathbf{W}}$ for each cell in $km^{-1}$. **b** The same for the null model, obtained by randomly reassigning directions to $\overrightarrow{\mathbf{W}}$ in each cell. In both cases, the $x$- and $y$-axis represent the Easting and Northing of the local Mercator projection in kilometers from Waterloo Bridge

intervening opportunities and those based on gravity-like approaches. Here we have considered different variations of these models. In the case of the gravity model, the deterrence function can show either an exponential or a power-law decay with the distance. For the intervening opportunities, we have focused on the radiation model[31] and its nonlinear version[45]. Models can be classified as unconstrained if only require the masses in every cell $m_i$ as inputs and production-constrained if additionally need the empirical outflow from each cell in order to estimate flows to other cells. The results discussed in this main paper refer to the unconstrained gravity with an exponential deterrence function and to the radiation model that is production-constrained. For the gravity model, the unconstrained version is considered because of its simplicity and amenability to analytical treatment (see Supplementary Note 4, Supplementary Figs. 14–19). The model parameters (for the gravity $k$ and $d_0$) have been adjusted to best reproduce the curve of the flux as a function of distance from the city center in terms of $R_P^2$. For the results of other models and details on the parameter calibration see Supplementary Note 3, Supplementary Tables 3 and 4, Supplementary Fig. 13, and Supplementary Note 6 along with Supplementary Figs. 23–26.

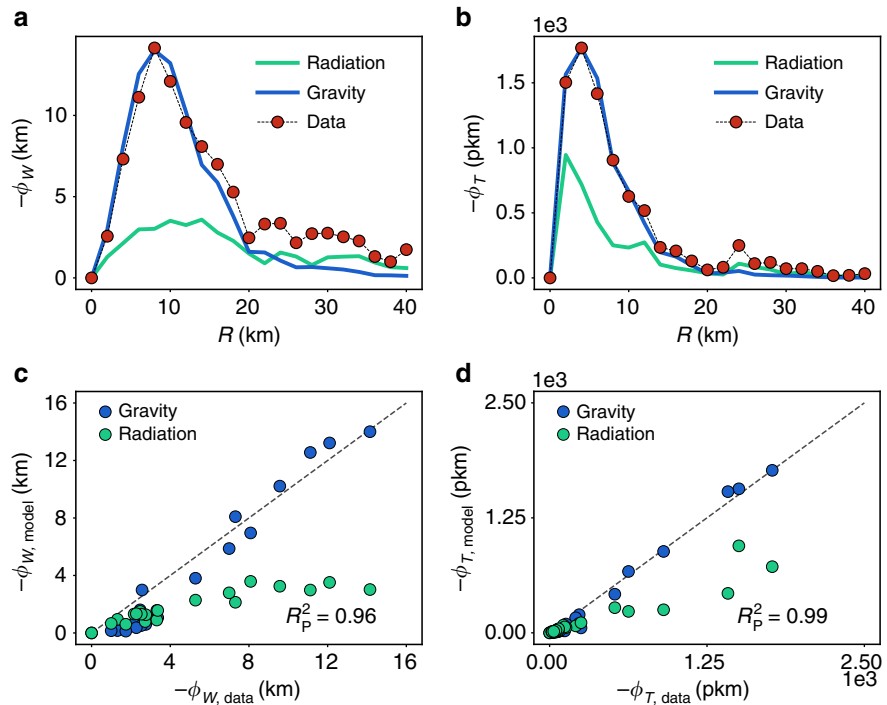

**Fig. 4** Model comparison. Red dots represent the fluxes measured with the empirical vector field. In the blue solid line, we show the predicted fluxes with the gravity model with an exponential deterrence function while the green curve corresponds to the radiation results. Results for London of **a** $\Phi_W$ with a comparison between the gravity model and the data of $R_P^2 = 0.96$ and **b** for the flux of T with $R_P^2 = 0.99$. In both cases, $d_0 = 9.4$ km. **c**, **d** are correlation plots between the gravity and the empirical flux. The results for the radiation are systematically below the diagonal. The units for $\Phi_W$ are in kilometers (km), while those of $\Phi_T$ are in persons multiplied by kilometer (pkm)

We consider a set of circles centered at the center of London with radius $R$ from 0 to 40 km (Supplementary Table 1). The flux of $\overrightarrow{W}$ across the circles with different $R$ is computed for both models and compared with the empirical value (Fig. 4). While the gravity model with an exponential deterrence function works well at reproducing the entering fluxes of the vector field $\overrightarrow{T}$ and $\overrightarrow{W}$ in the Greater London Area, the radiation model does not capture the level of fluxes observed empirically, despite receiving more detailed input information given that it is a production-constrained model. This is due to the fact that the local individual mobility predicted by the radiation model is more isotropic than the empirical one and the mobility predicted by the gravity. The results for other cities are consistent (Supplementary Note 7, Supplementary Figs. 27–42). The nonlinear radiation model improves a little the situation but it still underestimates the fluxes (Supplementary Note 6, Supplementary Figs. 23–26). The gravity with a power-law decaying deterrence function is neither able to reproduce well $\Phi_W(R)$ or $\Phi_T(R)$ (Supplementary Note 5, Supplementary Figs. 19–22). The unconstrained gravity framework provides the important advantage of allowing an analytical treatment for the fluxes, which is based on a scaling approach that is exact for the power-law deterrence function and approximated for the exponential (Supplementary Note 5 and Supplementary Fig. 22).

A recent brute-force comparison between models (gravity, radiation and intervening opportunities with different levels of contraint) and empirical commuting flows was carried out in[46]. The performance indicators at single flow level were favoring the exponential gravity model but the metrics were not able to capture big differences across models. For completeness, a similar analysis based on trip distance distribution has been included in Supplementary Note 10 and Supplementary Fig. 44. As with the direct flows, the results are not conclusive regarding model performance. However, the behavior of the fluxes as a function of the radius clearly discern between models performance. One may wonder what the origin is of these differences. The answer reveals the real potential of the vectorial framework. Besides the modulus, the empirical vectors $\overrightarrow{W}_i$ also have a direction that must be reproduced by the models. Measuring the angle of the vector over the horizontal positive axis $\Theta_{emp}$ and comparing it with the models predictions $\Theta_{mod}$, we obtain the scatter plots of Fig. 5 for London and Paris (results for other cities are in Supplementary Fig. 47). The domain of $\Theta_{mod}$ has been adjusted to minimize the difference. As seen in Fig. 5, the gravity model reproduces much better the direction of the vectors. Since the calculation of the fluxes involves a scalar product between $\overrightarrow{W}$ and the perimeter normal vector, the directionality (besides the modulus) is essential to obtain a good result. An analysis performed with direct trip flows would never be able to detect these differences.

**City potential.** Since we have empirically found that the field $\overrightarrow{W}$ can be considered irrotational, we can define a scalar potential using the formula $\overrightarrow{W} = -\nabla V$. Numerically, this means to find $V_i$ in every cell $i$ given the vector field $\overrightarrow{W}_i$. The procedure to do this is detailed in the Methods Section. Figure 6a shows the empirical potential for London obtained with Eqs. (12) and (13) compared with the one computed by the gravity model with exponential deterrence function using the same treatment as in Fig. 6b. The same results for Paris are displayed in Fig. 6d, e. Even though the empirical potential is noisier than the one obtained with the gravity model, they agree well. As shown in Fig. 6c, the level of correlation is $R_P^2 = 0.98$ for London and $R_P^2 = 0.93$ for Paris (Fig. 6f). The potential has a

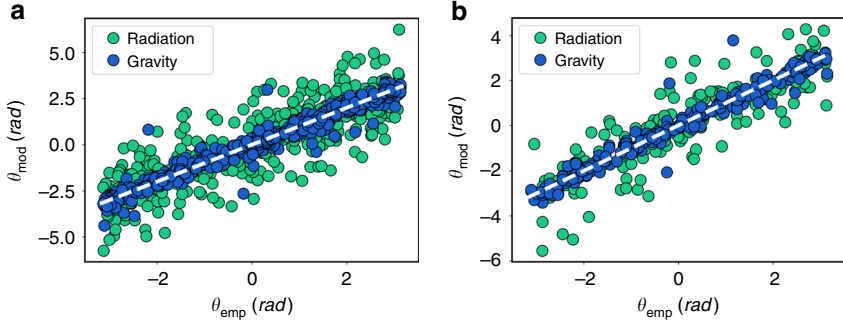

**Fig. 5** Angle comparison. Scatter plot of the angle of $\vec{W}_i$ in each cell $i$ respect to the positive horizontal axis measured from the data $\Theta_{emp}$ and compared with the models prediction $\Theta_{mod}$. The gray dashed lines correspond to the diagonals. The domain of the empirical angles is $(-\pi, \pi]$, while for $\Theta_{mod}$ we seek to minimize the distance to the empirical value by considering the original angle and its shifts in $\pm 2\pi$. In **a**, the comparison is performed in London and in **b** it is for the Paris case. R-squares for London are $R_P^2(\text{gravity}) = 0.96$ and $R_P^2(\text{radiation}) = 0.70$. For Paris, they are $R_P^2(\text{gravity}) = 0.96$ and $R_P^2(\text{radiation}) = 0.80$

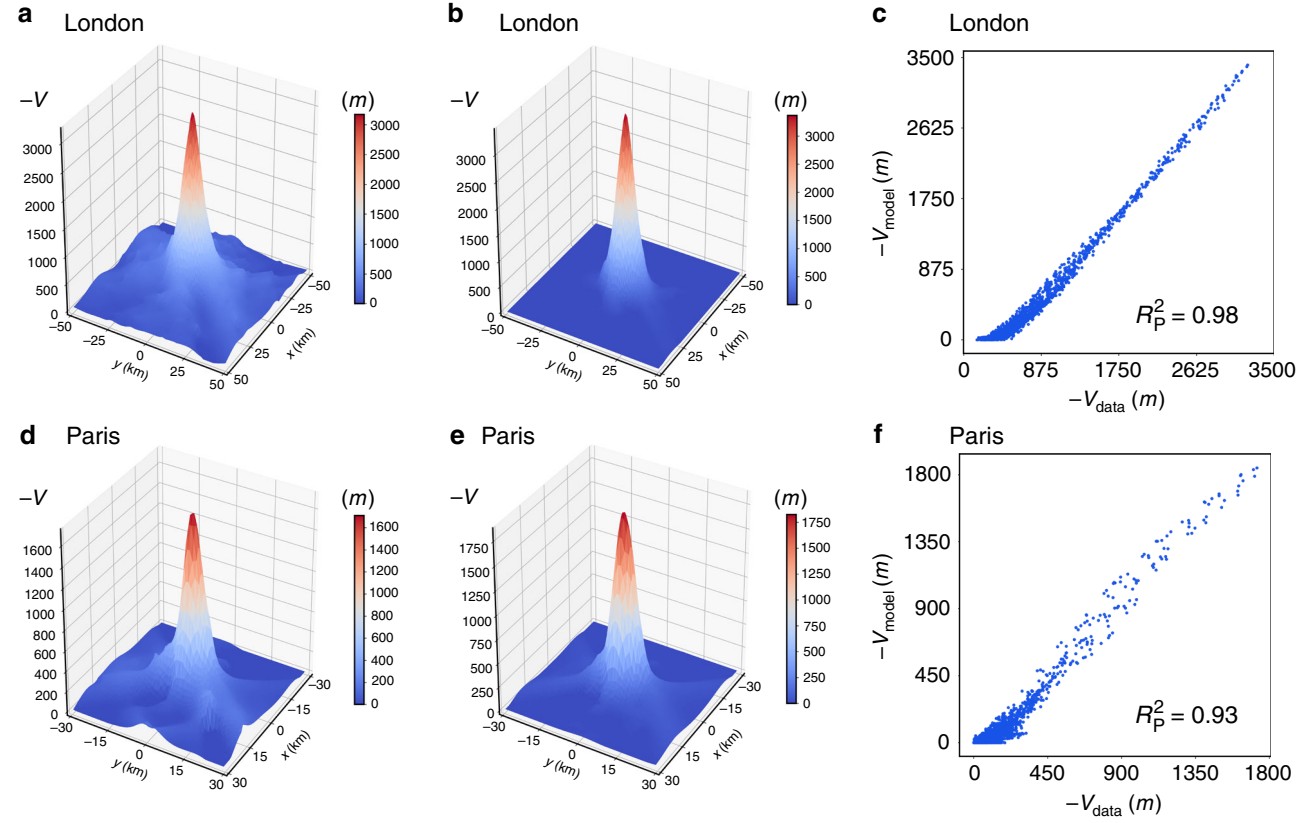

**Fig. 6** City potentials. **a–c** London. **d–f** Paris. **a, d** The empirical potential results clearly peaked in the city center, where in overall the density of inhabitants is high. The equilibrium point of the mobility is located at the minimum of the potential. **b, e** The gravity model predicted potential peaks also at the city center in agreement with the empirical results. **c, f** Scatter plots comparing gravity model with exponential deterrence function predictions and empirical values of the potential, which show high correlation

clear marked minimum in the center of the city, which is a clue of the commuting monocentricity at these scales. As depicted in Fig. 7, other cities or conurbations have a different configuration with as many local minima as mobility centers. Note that this is an appropriate method to define and visualize areas of attraction of each city and their geographic limits. The equipotential contour plots for other cities are shown in the Supplementary Note 8 and Supplementary Fig. 42.

## Discussion

In summary, we have introduced a vectorial field framework to characterize human mobility flows. When considering recurrent home-work mobility in cities, we find that the mesoscopic field representing the flows is well-behaved in the sense of satisfying Gauss's theorem and, besides, it is irrotational. As a consequence of this last point, it is possible to define a scalar potential, which reducing the dimensionality of the system encodes all the information on the commuting at a mesoscopic scale. The results are corroborated using two independent data sources for the commuting. Twitter data is used in the main text, and the results are reproduced for census data in the Supplementary Note 2 and Supplementary Figs. 2–10 for London, Manchester and Paris. Our focus here has been on commuting, which in most cities corresponds to over 60% of the total mobility. However, we cannot

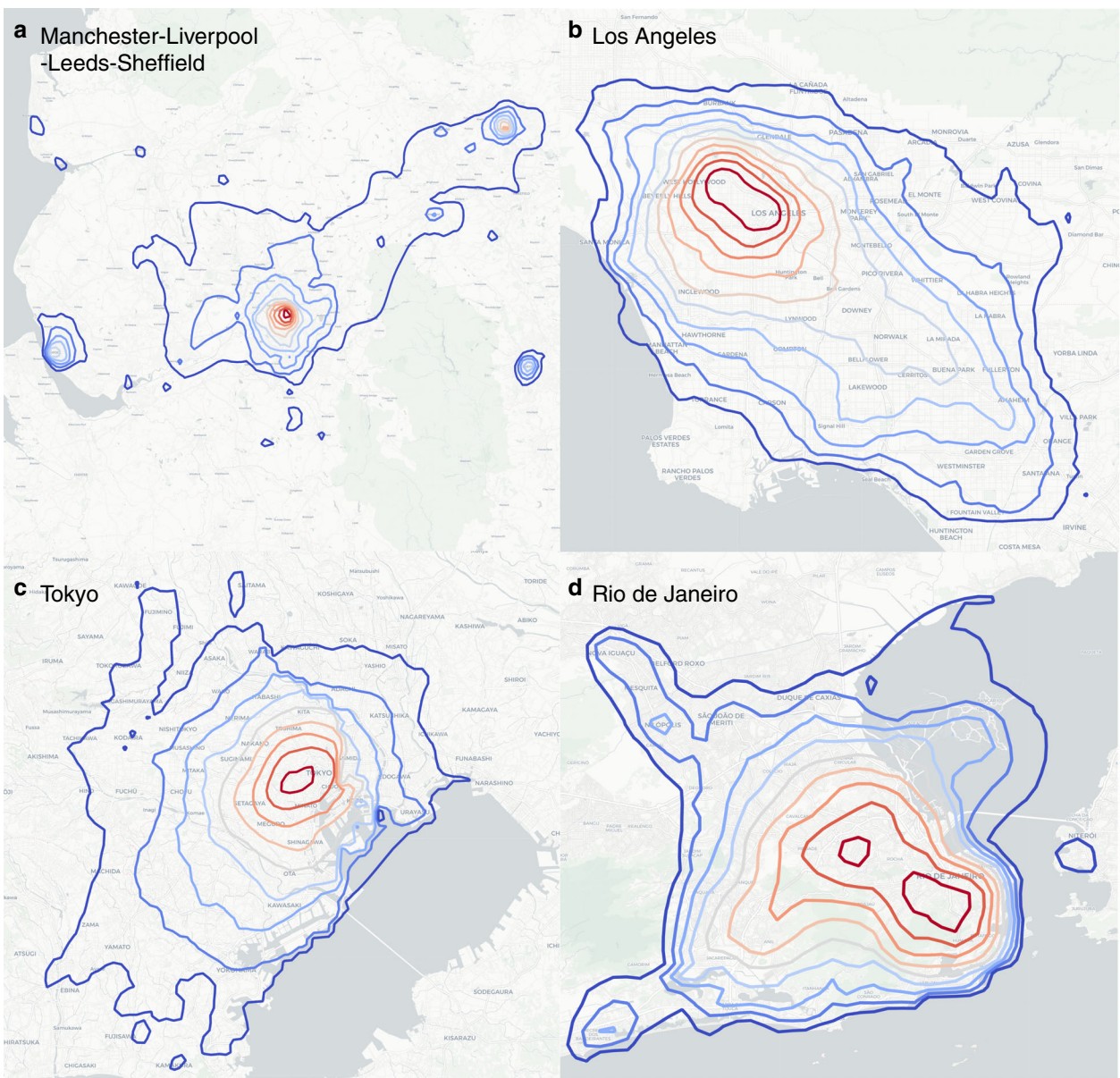

**Fig. 7** Empirical equipotential curves. Equipotential curves calculated with commuting flows obtained from Twitter data for several world cities and conurbations. **a** Manchester–Liverpool–Leeds–Sheffield (UK), **b** Los Angeles (USA), **c** Tokyo (Japan), and **d** Rio de Janeiro (Brazil). The underground map layout is produced using Carto. Map tiles by Carto, under CC BY 3.0. Data by OpenStreetMap, under ODbL

discard that other types of mobility at larger or shorter ranges may display similar behaviors. This remains as an open question for further exploration.

Our results have important consequences both from theoretical and applied perspectives. From a theoretical point of view, there are no a-priori reasons to assume that individual mobility at microscopic scale could induce a well-behaved mesoscopic field amenable to continuous treatment. Finding it from the empirical data implies that recurrent mobility in cities obeys deep symmetries that can be fully understood and described only within the framework of field theory. In particular, Gauss's and the rotational are the most basic theorems in the theory. They are the blocks upon which more involved results (metrics, theorems, etc) are built and this is why it is so important to prove that the vectors obtained from empirical data satisfy both. Gauss's

theorem means that the field is generated by a source and that the fluxes through surfaces must respect conservation laws. These constraints affect the flows and also the directions as shown in Figs. 4 and 5. The irrotational nature of the field implies that one can derive the field from a potential and vice versa, the field is univocally determined by the potential. The symmetries of the potential are also present in the field and, among other things, the dimensionality of the problem can be reduced: from a vector in every location to a scalar. Differences in the potential between points decide the direction and intensity of the mobility flows. Out of the symmetries usually it is possible to define invariant (conservative) quantities that play a central role in the vector field. Our work opens thus the door to use the heavy mathematical machinery developed during centuries to cope with vector fields.

Concentrating in the data, this framework allows to better distinguish between models performance. Any model trying to reproduce daily mobility flows should generate a field with the properties observed here in the empirical data. Otherwise, the model does not adjust to reality. These models have been used for decades to calculate trip demand in the planning of transport infrastructure. This is, therefore, a very relevant applied question. Recent brute-force comparisons between models and empirical commuting flows throw no clear conclusion on which model reproduces best the data. The metrics used were based on the analysis of raw mobility flows, hence a different approach is needed to reach a final conclusion. This is the role that the field theoretical perspective covers. Beyond the raw flows, the vector field has also a direction in each point and we can compare directions between model predictions and empirical data. This analysis shows that the gravity model with an exponential decay best reproduces both flows and directions. This result is further confirmed with the study of the fluxes across surfaces where the directionality plays a central role. We observe a better fit to the empirical curves as a function of the distance from the city center by the gravity model. Furthermore, the unconstrained gravity model admits an analytical treatment capable of producing expressions for the flux and the potential. This example is a proof of the potential of the vector representation.

In the gravity model framework, the existence of a potential has been postulated decades ago but these hypotheses were not systematically validated against data. We perform such validation and confirm that the gravity model with an exponential deterrence function generates a potential compatible with the empirical one. The potential is a fundamental tool to tackle hard open problems such as the definition of centers in cities, polycentricity and borders in conurbation systems. The shape of the potential sheds new light on the spatial organization of mobility in cities as we can picture city centers as the strongest gravitational attractors of the metropolitan area and redefine city boundaries. For example, borders could be defined as the locations where the potential falls below a fixed percentage from the highest peak of the city, separating thus the basins of attraction of the different centers. This can have an important practical relevance when planning infrastructures and public services.

## Methods

**Twitter data**. We use geolocated Twitter data in big cities and conurbations to extract information on commuters mobility. Even if the number of users is smaller than the local population, it has been shown that this data is valid to study aggregated urban mobility at scales larger than 1 km$^2$ with a global coverage[18,20]. Details on the procedure to download geolocated Twitter data are included in Supplementary Note 12. Our database is composed of tweets with coordinates in the area of Manchester–Liverpool, London, Los Angeles, Paris, Rio de Janeiro and Tokyo from March 2015 to October 2017. The information is then mapped into a regular square grid of 1 km$^2$. Tweets on Saturdays and Sundays, people moving faster than 200 km/h, users tweeting more than once per second, people tweeting <10 times in the whole time window and for less than one month have been filtered out. We consider the interval from 8 AM to 8 PM in local time as working hours, tweets in this interval are supposedly posted from the work place. Similarly, the rest of tweets are assumed to be posted from home. We assign to every user a home and a work cell as the most common cells during the corresponding hours. With this information, we can assume a daily trip from home to work for every user and another one back. Aggregating trips we can generate an OD matrix for the whole city, where each element $T_{ij}$ contains the number of people commuting from cell $i$ to $j$. The OD matrices represent generic levels of daily mobility and are used to determine trip demand for urban planning. The trips are not assigned to a particular moment in the data time window. To avoid noise due to poor statistics, we filtered out cells with <5 people as residents or workers.

A minor issue can raise with the misclassification of night-shift workers. A possible solution tested in[50] is to assume that the place with largest activity corresponds to work. However, this procedure was designed for more exhaustive data such as mobile phone records and it may introduce new biases with Twitter data. Still, the fraction of night-workers is only 10% of the total workforce in

London, and less than 11% in the whole UK (see [https://www.tuc.org.uk/news/260000-more-people-working-night-past-five-years-finds-tuc] for more details). The night workers mobility, even if misclassified, is part of the general daily mobility flow of the city. Finally, the census data is free from this issue since the questionnaire explicitly asks for residence and working places and the results are consistent for both data sources.

**Census data**. In addition to the Twitter data, the same study is repeated with census data from France and the United Kingdom. This data is publicly available on governmental web sites (FR, https://www.insee.fr and UK, https://www.ons.gov.uk/census/2011census). Census output areas have heterogeneous shapes different for every country and they do not compose a regular grid. A further treatment has to be carried out to adapt the population distribution and the home-work OD matrix to the grid. This introduces uncertainty that is not present in the Twitter data. Detailed information on how to divide and rearrange heterogeneous census areas into a square grid is provided in the Supplementary Note 5 and Supplementary Fig. 12. Thresholds on number of inhabitants and workers have been applied as well to avoid considering non statistically relevant zones. A method to assign a threshold to each city is provided in the Supplementary Note 2 and Supplementary Fig 11.

**Numerical calculation of the curl**. Given a vector field evaluated in the cells of a grid, it is possible to calculate the curl using the central finite differences[51] discretization method. The curl of $\overrightarrow{W}$ in the cell $i$, whose indices in the $x$- and $y$-directions are $(\alpha, \beta)$, is determined as:

$$\nabla \times \overrightarrow{W}_i = \frac{Wy_{(\alpha+1,\beta)} - Wy_{(\alpha-1,\beta)}}{2\Delta x} - \frac{Wx_{(\alpha,\beta+1)} - Wx_{(\alpha,\beta-1)}}{2\Delta y},$$

(3)

where $\Delta x$ and $\Delta y$ are side sizes of the cells in the $x$- and $y$-directions, and $Wx$ and $Wy$ are the $x$ and $y$ components of the vector $\overrightarrow{W}$, respectively, evaluated in $i$ and its nearest neighbors in the grid. The curl only has component in the $z$-direction since the vector $\overrightarrow{W}$ lays on the $x$–$y$ plane.

**Numerical calculation of the flux**. The definition of the flux as a perimeter (surface) integral is

$$\Phi_W^S = \oint_S \overrightarrow{W} \, \overrightarrow{n} \, d\ell$$

(4)

for the vector $\overrightarrow{W}$ and

$$\Phi_T^S = \oint_S \overrightarrow{T} \, \overrightarrow{n} \, d\ell$$

(5)

for $\overrightarrow{T}$. In both cases, the integral is performed over the perimeter $S$, $d\ell$ is the infinitesimal element of length and $\overrightarrow{n}$ is the unit vector normal to the perimeter in each point.

From a numerical perspective, the integrals are calculated as

$$\Phi_W^S = \sum_{i \in S} \overrightarrow{W}_i \, \overrightarrow{n}_i \, d\ell,$$

(6)

$$\Phi_T^S = \sum_{i \in S} \overrightarrow{T}_i \, \overrightarrow{n}_i \, d\ell,$$

(7)

where the index $i$ runs over all the cells intersecting the perimeter $S$, $\overrightarrow{n}_i$ is the unit vector normal to the surface in $i$ and $d\ell$ is approximated by the total perimeter of $S$ divided by the number of intersecting cells. The flux as a volume integral of the divergence is calculated as

$$\Phi_W^V = \sum_{i \in V} \left( \frac{Wx_{(\alpha+1,\beta)} - Wx_{(\alpha,\beta)}}{\Delta x} + \frac{Wy_{(\alpha,\beta+1)} - Wy_{(\alpha,\beta)}}{\Delta y} \right) dV$$

(8)

with the location of cell $i$ in $(\alpha, \beta)$, as above, the index $i$ runs over the cells in the volume $V$ and $dV$ is the area of the unit cell. The cells without resident commuters, $m = 0$, do not exhibit outflows and, to avoid inconsistencies, the field is defined as null in them. This implies that they do not contribute to the calculation of the flux or other results. Note that this is different from the classical continuous approaches of field theory in physics (e.g., electric or gravitational fields) where the field is defined everywhere and always contributes to the net flux.

**Gravity model**. The equation for the flow of commuters between two areas $i$ and $j$ with an exponential deterrence function is

$$T_{ij} = k \, m_i m_j e^{-d_{ij}/d_0},$$

(9)

where $k$ is a constant, $m_{i,j}$ are the populations of origin and destination areas $i$ ($j$), $d_{ij}$ is the distance between them and $d_0$ is a characteristic distance. This is the linear version of the Gravity Model, where the output and input flows are proportional to the number of people in the area. The model has only two parameters to fit ($k$ and $d_0$). The vector field is obtained by summing over the possible destinations and

dividing by $m_i$. If $\overrightarrow{\mathbf{u}}_{ij}$ is the unit vector pointing from $i$ to $j$, the vector field can be written as

$$\overrightarrow{\mathbf{W}}_i = \sum_j \frac{T_{ij}}{m_i} \overrightarrow{\mathbf{u}}_{ij} = k \sum_j m_j e^{-d_{ij}/d_0} \overrightarrow{\mathbf{u}}_{ij}. \tag{10}$$

**Radiation model**. The Radiation Model is inspired by radiation and absorption of particles[31]: for every worker residing in and leaving cell $i$, the destination (work) cell $j$ is obtained using the probability expression

$$P(i,j) = \frac{m_i m_j}{(m_i + s_{ij})(m_i + m_j + s_{ij})}, \tag{11}$$

where $s_{ij}$ is the population residing in a circle centered in $i$, with radius $d_{ij}$ and excluding the populations of $i$ and $j$. The average flows can be calculated as $\langle T_{ij} \rangle = T_i P(i,j)$, where $T_i$ is the empirical total outflow of cell $i$.

**Numerical calculation of the potential**. The potential is calculated by numerically solving the equations $-\nabla V_i = \overrightarrow{\mathbf{W}}_i$ taking into account that $\nabla \times \overrightarrow{\mathbf{W}} = 0$. For the computation of the empirical potential, we used conditions $V = 0$ in all the boundary regions of the grid and then use the forward centered discretization formula for the gradient operator[51] starting from the city bounding box corner. In a cell $i$ with indices $(\alpha, \beta)$, this operation becomes:

$$\frac{dV_i}{dx} = \frac{V_{\alpha+1,\beta} - V_{\alpha,\beta}}{\Delta x} = W_{(x),\alpha,\beta}, \tag{12}$$

$$\frac{dV_i}{dy} = \frac{V_{\alpha,\beta+1} - V_{\alpha,\beta}}{\Delta y} = W_{(y),\alpha,\beta}, \tag{13}$$

The procedure is iterated until all cells have been assigned a potential. We average then the resulting potentials after starting from every corner of the bounding box to decrease the noise.

## Data availability

In this work, we use two data sources: Geolocated Twitter and census in the UK and France. All the data are available online, although in all cases the access conditions require the user to obtain the data directly from the provider sites. For the census data, the 2011 UK commuting information can be found at output area level in the link [https://wicid.ukdataservice.ac.uk/cider/about/data_int.php?type=2] and 2011 French data at municipal level is available at [https://www.insee.fr/en/statistiques?categorie=1]. For Twitter, the data is downloaded using the streaming API [https://developer.twitter.com/en/docs/tweets/filter-realtime/overview]. An example of the script employed to obtain geolocated data in a geographical area is provided in the Supplementary Note 12. The aggregated information necessary to reproduce our results has been uploaded at the repository Figshare with doi: [https://doi.org/10.6084/m9.figshare.8158958][52].

## Code availability

An example of the code used to collect Twitter data is provided in the Supplementary Note 12. The code for the analysis was programmed using Python and the equations employed are described in the Methods Section.

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

## Acknowledgements

M.M. and A.B. are funded by the Conselleria d'Innovació, Recerca i Turisme of the Government of the Balearic Islands and the European Social Fund. M.M., A.B., P.C., and J.J.R. also acknowledge partial funding from the Spanish Ministry of Science, Innovation and Universities, the National Agency for Research Funding AEI and FEDER (EU) under the grants ESOTECOS (FIS2015-63628-C2-1-R and FIS2015-63628-C2-2-R) and PACSS (RTI2018-093732-B-C22) and the Maria de Maeztu program for Units of Excellence in R&D (MDM-2017-0711). M.L. received financial support from a grant of the French National Research Agency (project NetCost, ANR-17-CE03-0003).

## Author contributions

M.M., A.M. and J.J.R. designed the study and contributed new conceptual tools. M.M., A. B. and M.L. cleaned and processed the data. M.M. and A.M. performed the numerical analyses. M.M. and J.J.R. developed the analytical treatment. P.C. and J.J.R. coordinated the study. All authors contributed to the discussion, to the writing and approved the paper.

## Additional information

**Competing interests:** The authors declare no competing interests.

