## [Peer Review File · Nature Communications]

Reviewer #1 (Remarks to the Author):

I thought this work was interesting and novel. I'm not so sure how useful the result will be, though the authors offer some good suggestions. However, it is useful to add this approach to the tools available when studying mobility data.

I have a few points the authors might want to consider and possibly to comment on in the text. The only point I feel is required before publication is regarding the source of, and availability of, the data used here.

I did not see the source of these data sets referenced. I feel that should be done here. The data used in this paper should be publicly available in some form.

The fluxes are defined in an asymmetric way in eqn 1. You appear to define your fluxes as commuters travelling from home to work though this is not made clear when you define T_{ij} . Assuming that is your approach, could you do this the other way round? Indeed night workers will be encoded in your twitter data the wrong way round given you use normal working hours to define home and work for the twitter data.

What is the definition of the "mass" parameters used in each case? You refer to "local population" (after equation 1) which is well defined in the census data. However what population? Perhaps this is the population as given in a census, rescaled as in SI equation (1). Is the same mass used for the Twitter data or do you use a different measure derived from the Twitter data or another Twitter data stream? I would guess having defined the home location of a twitter user and then you sum those to use as the mass in the twitter case. Why not do the same for the census data? That is the census uses the reported population, the other seems to use the actual flows in the twitter data.

What would be the interpretation of a significant non-zero rotational element to the field? As far as I can see on any scale larger than a block of one city are commuter flows are unlikely to be rotational. I have seen a one-way system on the scale of a block in many cities (e.g. New York), even some one way loops on metro systems (e.g. at Heathrow, London or on line 10 of the Paris metro) but these are exceptional. Really it is natural to have no rotation in this context and hence I would suggest the existence of a potential is natural.

I find it very odd that the unconstrained gravity model was used. After all, adding the input and output constraints is trivial numerically these days. The radiation model has an advantage (!) with its output constraint. However, your results speak for themselves. Is there anything more to comment on regarding this?

I did not see a note on the significance of the $\gamma=1$ case in two spatial dimensions. Really it is the converse, perhaps it worth noting what special property your mobility field and potential does not have because fluxes are not $1/r$ dependent. For instance, maybe highlight the dimensions of the space you are working in (two) when you mention $1/r$ as the appropriate field in section 5 of SI. You said (in sec 5 of SI) "we are going to consider the case of $\gamma=1$ " but I did not see that in the analytical work or in figs S20 and S21.

Tim Evans

Reviewer #2 (Remarks to the Author):

The manuscript presents a general and detailed characterization of the mobility flow in cities in terms of a vector field. The field is shown to be irrotational and to satisfy --approximately-- Gauss'

divergence theorem. The authors successfully link the empirically constructed vector field to the one derived from the so-called gravitational model, a well accepted model for the microscopic dynamics of human mobility.

I think the work is exhaustive, statistically well conducted and provides interesting and new results. I guess that the reproducibility is guaranteed, although this is something one can only claim after trying to reproduce in detail the whole amount of results, which of course I've not been able to do. From the methodological side, it is very valuable to provide a systematic method to characterize the general properties of the mobility flows, done in the paper through the construction of the vector field. In addition, the comparison to the vector field constructed upon the results provided by the models somehow highlights the suitability of the gravity model in front of other models of mobility. After reading the manuscript several times, however, I somehow miss the message that wants to be conveyed. I would therefore suggest the authors some clarifications that I am sure would improve the readability and scope of their manuscript.

The main issue I find is the following: The authors claim that the irrotational nature of the vector field is a signature of non-trivial flow organization. However, they do not provide clues on how or why could it be different. I guess that means that the flows in a given planar coordinate --let's say, the angle-- are balanced, whereas in the other coordinate --let's say, the radius-- are not? How could it be different? In addition, the authors should explicit --in the main manuscript:

- i) That the observed flows belong to a certain time interval, otherwise there are serious flow continuity issues,
- ii) That the links T_{ij} are among adjacent cells in the grid: this point is not made clear and can induce to confusion, because a trip can be from one side of the side to the other, and, consequently, no itinerary information is provided and no vector field can be extracted --it could be that I missed some important detail, in that case, I urge the authors to explain the construction of the vector field better.
- iii) Stating Gauss' theorem in a single equation in the main manuscript would help to visually understand what is being computed to the non-specialized audience.

I think answering these questions is relevant to grasp the strength of the provided results.

I finally have a personal suggestion that the authors may consider. I think it would be interesting to know how the provided results link to well established facts concerning 'gradient networks'. The construction of the vector field reminds me a lot to the construction of a gradient network, a concept defined in the framework of mobility, but completely general. There are several facts that suggest that a gradient network can be at least similar to the proposed framework. By construction, if one extracts of vector field from a gradient network that will be irrotational. In addition, one can define basins of attraction and, consequently, a potential for the vector field. Finally, some interesting properties, in terms of flows, can be inferred from gradient networks, which may enrich the amount of nice results provided by the authors. For further details, take a look to the references below:

Z Toroczkai, KE Bassler (2004)
Jamming is limited in scale-free systems
Nature 428 (6984), 716

Z Toroczkai, B Kozma, KE Bassler, NW Hengartner, G Korniss (2008)
Gradient networks
Journal of Physics A: Mathematical and Theoretical 41 (15), 155103

Connecting these two approaches would certainly improve the generality of the manuscript.

I think the paper has potential enough to be published in Nat Comms, after addressing the issues I mentioned. I will be happy to read the revised version of the manuscript.

With best regards,

Bernat Corominas-Murtra

Reviewer #3 (Remarks to the Author):

In this manuscript, the authors proposed a novel methodology to analyze daily commuting data. The authors first made a comprehensive literature review in the area of human mobility. The method for defining the vector field is new and interesting. Twitter data and census data are used to obtain the empirical results under the framework of field theory. Predicted fluxes of gravity models and radiation models were compared with the fluxes measured with the vector field. Overall, the paper is well organized and written. However, I think the paper is too concise that some necessary information is not included.

Major comments:

1. The proposed vector field method is new and interesting, however, the advantage to use vector field for mapping daily commuting flows is not clear to me. I would be glad to see how the vector field method can improve the performance of mobility analysis, gravity models or radiation models. A comparison with previous methods for mapping commuting flows is suggested.
2. The contribution of the work to the field of human mobility needs further elaboration and explanation. The strength of the proposed vector field method needs to be further elaborated.
3. The mean outgoing mobility direction of each cell can be measured using the defined resultant vector. The variance of outgoing mobility direction may be also interesting and encapsulates additional characteristics of outgoing mobility flows. I suggest the authors analyze the variance of outgoing mobility direction for different cells. Some interesting findings may be discovered.
4. Does T_{ij} in Eq. (1) represent daily flow or hourly flow? Does the commuter vector field show different patterns during different time periods of a day?
5. The authors made great efforts to demonstrate that the field fulfills Gauss theorem and has irrotational character. Please elaborate why these two findings are important? Does the irrotational character of field only supports the definition of a potential? Can the two findings be connected to some actual phenomena of commuter flows? Could we obtain some insights in commuter flows based on the two findings?

Minor comments:

1. The last sentence of page 3: please elaborate why radiation model receive more detailed input information.
2. Please supplement a concrete definition of entering flux of vector field T in the main manuscript. This will help readers better understand Fig. 4 b and d.
3. Which model did the authors use to obtain the results in Fig. 5?

First of all, we would like to thank the reviewers for their comments that surely contribute to improve the manuscript.

Reviewer #1

I thought this work was interesting and novel. I'm not so sure how useful the result will be, though the authors offer some good suggestions. However, it is useful to add this approach to the tools available when studying mobility data.

We find in this work a feature of urban-scale daily mobility that has not been observed before: the fact that it can be encoded as a vector field. Related features such as the existence of a potential have been postulated decades ago within the framework of the gravity model but these hypothesis were never validated against empirical data. The implications of this are many fold. From a theoretical perspective, we are gaining new insights on empirical mobility systems and opening the door to use on them the very heavy mathematical machinery developed in Physics during centuries to cope with vector fields. We admit that this is mainly theoretical, but it is a necessary component in the search for knowledge. From a more applied perspective, any model trying to reproduce daily mobility flows should produce a field fulfilling the properties observed here. Therefore, we are providing a filter to validate or falsify flow models. Moreover, the potential described here can be a key to tackle hard problems such as the definition of centers in cities, polycentricity and borders in a conurbation system. For example, borders could be defined as a fixed percentage from the highest peak of the city potential separating the basins of attraction of the different centers. This has an important practical relevance when planning infrastructures and public services. This is still far away, much work is needed, but we are giving here a first step in this direction.

Still, this is a comment that several reviewers have done in different ways. Consequently, we have modified the conclusion section to include a more extensive discussion on it.

I have a few points the authors might want to consider and possibly to comment on in the text. The only point I feel is required before publication is regarding the source of, and availability of, the data used here.

There are two main data sources in this work: Twitter and census. In the new version of the manuscript and SI, we explain how to download the raw data including links for both sources and scripts for the Twitter API. Specifically, and following Nature policy, we have added now a "Data Availability" statement in the main manuscript detailing data accessibility, the corresponding links. An example of the code to do queries in the API streaming data is included in the Section 12 of the SI. Additionally, we have uploaded in the repository Figshare the aggregated information essential to reproduce our results. This data will be available upon the manuscript release and it has an associated DOI as indicated in Ref. [53]. Still, it can be already accessed with this private link: <https://figshare.com/s/9b5cc81af1693d311503>.

The fluxes are defined in an asymmetric way in eqn 1. You appear to define your fluxes as commuters traveling from home to work though this is not made clear when you define T_{ij} .

Thanks a lot for pointing this issue out, we have updated the definition of T_{ij} to be clearer and specifying that the flows are from home (cell i) to work (cell j).

Assuming that is your approach, could you do this the other way round? Indeed night workers will be encoded in your twitter data the wrong way round given you use normal working hours to define home and work for the twitter data.

This is an interesting but tough question. Inverting the flows, as long as they are summed and assigned to the residence place, reverses the direction of the fields but does not modify our results: The field takes in every point the opposite direction but the modules remain unchanged. The potential also changes sign but not functional shape. The attractive fields become repulsive but beyond signs nothing else gets altered. On the other hand, if the resultant vectors are summed and assigned to the working place, the field is slightly modified in every location but the mesoscopic organization neither changes. This case is now considered in SI Section 13.

Distinguishing night workers is rather complex, mostly because they move almost in inverse phase respect to the rest of the population. For most of the people, we are talking about 8 hours of work and 8 hours of sleep (likely at

home) as shown in time use surveys (https://ec.europa.eu/eurostat/cache/metadata/en/tus_esms.htm). One possibility is to check for each user the time period of highest activity and associate it to work. Although, this is not free of biases because the peak could be not only at work but, for example, before or after sleeping at home. A similar idea was applied to mobile phone records in our recent paper (Bassolas et al., Transportation Research Part A **121**, 56-74 (2019)) but with that type of data the information per user is more precise. Since this procedure is untested and requires further validation work, following Occam's razor we have taken the simplest approach. Fortunately, the error is small since only 10% of the total workforce in London works at night, and not more than 11% in the whole UK (<https://www.tuc.org.uk/news/260000-more-people-working-night-past-five-years-finds-tuc>). Moreover, the composition of the so-called shift-workers, is variable in time, i.e. they not only work at night, but many of them change their schedules from night work to day work from week to week (<http://www.hse.gov.uk/research/rrpdf/rr887.pdf> see Fig.6). 82% of workers never experience work shifts (<http://www.hse.gov.uk/research/rrpdf/rr887.pdf> Table 1). The night workers mobility, even if misclassified, is part of the general mobility flow of the city.

Finally, it is important to note that the census data is free from this issue since the census questionnaire explicitly asks for residence and working places and, as we show in the manuscript, the results are consistent for both data sources. Still this is an important question and we have added a detailed discussion in the Methods section (Twitter data) when describing the data collection and filtering.

What is the definition of the "mass" parameters used in each case? You refer to "local population" (after equation 1) which is well defined in the census data. However what population? Perhaps this is the population as given in a census, rescaled as in SI equation (1). Is the same mass used for the Twitter data or do you use a different measure derived from the Twitter data or another Twitter data stream? I would guess having defined the home location of a twitter user and then you sum those to use as the mass in the twitter case. Why not do the same for the census data? That is the census uses the reported population, the other seems to use the actual flows in the twitter data.

Thanks for pointing this out. In the model definition, we refer to the mass as "local population" in a generic way but this was confusing. What we are considering as mass is $m_i = \sum_j T_{ij}$ including $j = i$, for both datasets in a coherent way. This is an approximation of the total workforce in every cell. As we showed in Ref. [46], this definition of mass yields better flow estimates than the total population (for both gravity and radiation models) and it is easier to obtain from Twitter than the total population. We have modified the manuscript to make this point clear now (Sec. Results, final paragraph of the "Definition of the vector field").

What would be the interpretation of a significant non-zero rotational element to the field? As far as I can see on any scale larger than a block of one city are commuter flows are unlikely to be rotational. I have seen a one-way system on the scale of a block in many cities (e.g. New York), even some one way loops on metro systems (e.g. at Heathrow, London or on line 10 of the Paris metro) but these are exceptional. Really it is natural to have no rotation in this context and hence I would suggest the existence of a potential is natural.

This is a question that we have asked ourselves when we saw the results of the rotational analysis. Circular infrastructures are not so uncommon in cities, apart from the circular metro lines many highways are organized as concentric rings when there is no major geographical impediment like in Paris or London. Still we would need an unbalanced flow of people living in an area over the ring and working in another to find net rotation elements. An illustrative but unrealistic example is plotted in the following sketch:

Here we have one residential cell A, and two industrial areas B and C (with no population). If the number of workers attracted by B is larger than those of C, for instance because it offers more jobs, the net flow would produce a non-zero rotational element in the field. What we see is that this does not happen anywhere in the cities under study. Our guess is that the land use mixing in large cities is strong enough to prevent this sort of loops at mesoscopic scales, favoring a more hierarchical configuration of the field with a few clear centers. It may happen, however, far in the outskirts or in the country side, where the land use is more segregated. We studied land use mixing in Royal Society Open Science **2**, 150449 (2015), linking the mobility field and the land use could be an interesting topic for next works. The irrotational character of the field can be natural but it does not look trivial to us. We have added now a paragraph explaining this issue in Section Results, Empirical Results.

I find it very odd that the unconstrained gravity model was used. After all, adding the input and output constraints is trivial numerically these days. The radiation model has an advantage (!) with its output constraint. However, your results speak for themselves. Is there anything more to comment on regarding this?

Certainly, the unconstrained gravity strongly outperforms the radiation model. The data shows that a potential can be defined, and the gravity is a model well suited to generate one. We did not use the production constrained gravity model because the analytical treatment becomes rather complicated compared with the unconstrained version. The unconstrained model already reproduces the features observed in the empirical flows and it is much simpler to analyze. We remark this in the new version of the manuscript, in Section Results, subsection Models.

I did not see a note on the significance of the $\gamma=1$ case in two spatial dimensions. Really it is the converse, perhaps it worth noting what special property your mobility field and potential does not have because fluxes are not $1/r$ dependent. For instance, maybe highlight the dimensions of the space you are working in (two) when you mention $1/r$ as the appropriate field in section 5 of SI. You said (in sec 5 of SI) "we are going to consider the case of $\gamma=1$ " but I did not see that in the analytical work or in figs S20 and S21.

As the reviewer indicates, the case $\gamma = 1$ is very special since it corresponds to the gravitational and electrical fields in two dimensions and it deserves to be highlighted. We have added now a commentary in the section of the SI considering this case and a few sentences in the Sec. Methods, sub-Sec. Numerical calculation of the flux.

Reviewer #2

The manuscript presents a general and detailed characterization of the mobility flow in cities in terms of a vector field. The field is shown to be irrotational and to satisfy --approximately-- Gauss' divergence theorem. The authors successfully link the empirically constructed vector field to the one derived from the so-called gravitational model, a well accepted model for the microscopic dynamics of human mobility.

I think the work is exhaustive, statistically well conducted and provides interesting and new results. I guess that the reproducibility is guaranteed, although this is something one can only claim after trying to reproduce in detail the whole amount of results, which of course I've not been able to do. From the methodological side, it is very valuable to provide a systematic method to characterize the general properties of the mobility flows, done in the paper through the construction of the vector field. In addition, the comparison to the vector field constructed upon the results provided by the models somehow highlights the suitability of the gravity model in front of other models of mobility. After reading the manuscript several times, however, I somehow miss the message that wants to be conveyed. I would therefore suggest the authors some clarifications that I am sure would improve the readability and scope of their manuscript.

Thanks a lot for these positive comments. As we see it, the contribution of our work is relevant from two perspectives. One is that we find features in the mobility data that have important consequences from a theoretical point of view. On this sense, this is mostly advance of knowledge. It has as well applied implications such as those related to the definition of cities, boundaries and polycentricity. This means that further studies might develop metrics based on our approach, which opens the possibility of using the extensive set of tools

introduced in physics and mathematics to deal with vector fields. We have tried to highlight this better in the conclusions of the new version of the manuscript.

The main issue I find is the following: The authors claim that the irrotational nature of the vector field is a signature of non-trivial flow organization. However, they do not provide clues on how or why could it be different. I guess that means that the flows in a given planar coordinate --let's say, the angle-- are balanced, whereas in the other coordinate --let's say, the radius-- are not? How could it be different? In addition, the authors should explicit --in the main manuscript:

The main situation in which this can be different is when the land use is strongly segregated and mediates in the form of the mobility. A sketch with an example is provided in the answer to the reviewer 1. Mainly, rotation in the flows can emerge when working and residence areas are separated and when the attraction of a cell is not related to the population. This does not happen inside metropolises, because the land use is typically mixed and jobs related to services are attached to population. We have modified the manuscript to explain this point better in Section Results, last paragraph of the sub-Section Empirical results.

i) That the observed flows belong to a certain time interval, otherwise there are serious flow continuity issues,

The flows are estimated using two photographs of the cities: one at night (when we assume that people are at home) and one during working hours, in workdays from Monday to Friday. To obtain home and work for every user, we are aggregating Twitter data for two years and a half (mid 2015 to end 2017). The time period has been included in the Methods section on Twitter data. The most common place during the night hours is taken as home cell and, similarly, the one for working hours as work cell. Then we have created a picture of mobility in a standard working day, as in the census, summing the flows between home and work cells for all the users. This produces the OD matrices that we use in the analysis. We are not assigning this mobility to a particular time window during those two years, it is true that if the process is repeated later the users may change. For example, depending on the city it may have seasonal differences between summer and winter. However, at the mesoscopic scale of flows the information tends to be more stable than at individual level. The typical time scales for changes in city ODs is closer to the time window between census, decades instead of years. We have added a discussion on this in the Methods section of the manuscript, in the sub-Section on Twitter data.

ii) That the links T_{ij} are among adjacent cells in the grid: this point is not made clear and can induce to confusion, because a trip can be from one side of the side to the other, and, consequently, no itinerary information is provided and no vector field can be extracted --it could be that I missed some important detail, in that case, I urge the authors to explain the construction of the vector field better.

The flows T_{ij} can be also between non adjacent cells, but it is true that we include no information on the trajectories since we are working only with origin-destination mobility. The results could be different using trajectories, and it may be interesting for future works. This was also related to a question of another reviewer so the manuscript text was not very clear. We have now clarified the meaning of T_{ij} in the new version of the manuscript in Section Results, sub-Section Definition of the vector field.

iii) Stating Gauss' theorem in a single equation in the main manuscript would help to visually understand what is being computed to the non-specialized audience.

Thanks for this suggestion, we have added this information in the manuscript in the section on Empirical Results where we discuss Gauss's theorem.

I think answering these questions is relevant to grasp the strength of the provided results.

I finally have a personal suggestion that the authors may consider. I think it would be interesting to know how the provided results link to well established facts concerning 'gradient networks'. The construction of the vector field reminds me a lot to the construction of a gradient network, a concept defined in the framework of mobility, but completely general. There are several facts that suggest that a gradient network can be at least similar to the proposed framework. By construction, if one extracts of vector field from a gradient network that will be irrotational. In addition, one can define basins of attraction and, consequently, a potential for the vector field. Finally, some interesting properties, in terms of flows, can be inferred from gradient networks, which may enrich the amount of nice results

provided by the authors. For further details, take a look to the references below:

Z Toroczkai, KE Bassler (2004)
Jamming is limited in scale-free systems
Nature 428 (6984), 716

Z Toroczkai, B Kozma, KE Bassler, NW Hengartner, G Korniss (2008)
Gradient networks
Journal of Physics A: Mathematical and Theoretical 41 (15), 155103

Connecting these two approaches would certainly improve the generality of the manuscript.

Gradient networks are based on a scalar field defined in every node as it could be our potential. This formalism can be useful to uncover hidden symmetries in mobility and it looks promising. There are some issues to think around like whether the connections must follow a strict distance-base dependence or not. Still it can be a matter to consider. Here we are linking vector field theory and mobility that already represents a major leap. Thanks, however, for a nice suggestion that can be a promising future research venue.

Reviewer #3:

In this manuscript, the authors proposed a novel methodology to analyze daily commuting data. The authors first made a comprehensive literature review in the area of human mobility. The method for defining the vector field is new and interesting. Twitter data and census data are used to obtain the empirical results under the framework of field theory. Predicted fluxes of gravity models and radiation models were compared with the fluxes measured with the vector field. Overall, the paper is well organized and written. However, I think the paper is too concise that some necessary information is not included.

Major comments:

1. The proposed vector field method is new and interesting, however, the advantage to use vector field for mapping daily commuting flows is not clear to me. I would be glad to see how the vector field method can improve the performance of mobility analysis, gravity models or radiation models. A comparison with previous methods for mapping commuting flows is suggested.

We have performed a recent brute-force comparison between models (gravity, radiation and intervening opportunities with different constrain levels) and empirical commuting flows in Ref. [46]. The performance indicators at single flow level were favoring the exponential gravity model but the metrics were not able to capture big differences across models. This means that a completely different framework was needed to be able to extract information from the data and to understand which is the best approach in the literature regarding aggregated commuting flows. This is the role that the field theoretical perspective covers. For example, beyond the raw flows comparison, the vector field has also a direction in each point and we can compare directions between model predictions and empirical data. Following the reviewer's suggestion, we have performed this clarifying analysis and included it now in the manuscript in the last paragraph of the Section Results, sub-Section Models, Figure 5 and Figure S47. Here we see the full potential of the vector representation. While the performance of the models is similar at flow levels, the vector directions are way better reproduced by the gravity model. This is related to the differences in the fluxes because the flux compresses the modulus and the direction of the resultant mobility vectors.

Also as the reviewer suggested, we have included a comparison among models in Fig S44 using trip-length distributions, which is a standard in the characterization of mobility in surveys and census. As discussed in Refs. [44,46], the direct comparison shows that the gravity outperforms the radiation model in distances below the typical city size around 10 km as it was already known from the literature. This has been also mentioned in the manuscript in section Results, sub-Section Models. This metric, as those employed in Ref. [46], is not able to discern well between models performance even though we have included this analysis for the sake of a complete data description.

2. The contribution of the work to the field of human mobility needs further elaboration and explanation. The strength of the proposed vector field method needs to be further elaborated.

Thanks for this comment. We have re-elaborated the conclusion section to highlight better the advantages that the vector framework can bring to mobility analysis. This includes as an example the new analysis performed regarding mobility direction. The new formalism brings advances in two separate fronts: One is theoretical, given that we observe that a well behaved vector field can be defined from the empirical data. This corresponds to an intrinsic property of the data, which adds to the present knowledge on mobility. The derivations from this theoretical knowledge are beyond this work, but essentially the possibility of using all the vector field mathematical machinery developed in physics and mathematics in the last centuries is now open. From an applied perspective, we can develop new metrics based on the vectors. A simple direct example has been discussed in the previous answer, another example is the flux through surfaces but one could easily think about more complex metrics involving, for instance, the potential.

3. The mean outgoing mobility direction of each cell can be measured using the defined resultant vector. The variance of outgoing mobility direction may be also interesting and encapsulates additional characteristics of outgoing mobility flows. I suggest the authors analyze the variance of outgoing mobility direction for different cells. Some interesting findings may be discovered.

We find this very interesting and inspired us for the analysis on the angle comparison between models and empirical data. We added Section 11 of the SI on angle distributions and discuss some of the results in the main text at Section Results, sub-Sections Empirical Results and Models. We have performed the analysis in two steps: The first one, intended to confirm the irrotational nature of the vector field, consists in studying the angle distribution of the resultant vectors. We have computed the direct distributions and their corresponding complementary cumulative functions in London, Paris and Los Angeles. The distributions are flat and the complementary cumulative functions are linear, confirming that the distribution is uniform and compatible with an irrotational field (Figures S45 and S46).

The second analysis focuses on the directions (angles) of the vectors contributing to each resultant vector as suggested by the reviewer. In each cell, we compute the average of the absolute value of the angle differences between every pair of vectors \vec{T}_{ij} constituting \vec{T}_i and call it $\langle \Delta \rangle$. The spread of values of $\langle \Delta \rangle$ informs on the directional heterogeneity of the flows departing from each cell. The distribution of $\langle \Delta \rangle$ values is similar across the three cities considered (Los Angeles, London and Paris) and it is quite uniform in space, except for a few outliers. When compared with a null model in which the vector directions are taken at random in each cell, we find that the empirical $\langle \Delta \rangle$ is much lower, indicating that the directions of the vectors are concentrated. We have added a description of the analysis in Section 11 of the SI including Figures S48, S49 and S50.

4. Does T_{ij} in Eq. (1) represent daily flow or hourly flow? Does the commuter vector field show different patterns during different time periods of a day?

The commuting patterns are considered as daily mobility. In the census questionnaires, the residence and the working areas are requested for each individual. A trip in each direction is assumed every working day and the commuting flows are calculated aggregating individuals living and working in the same areas. In the case of the Twitter data, we reproduce this information by detecting the most frequent zone from which users tweet at working and non-working hours. Then we aggregate the users to obtain an approximation to commuting flows. The analysis of the vector field is carried out with only one of the two trips, in this case from home to work. The results with the opposite trip are equal but changing the vectors to the opposite direction and the signs of the gravity model and the potential. As this was also a question of other reviewers, we have clarified this point in the manuscript text (Methods Section, sub-Section Twitter data and in the definition of T_{ij} in the Section Results, sub-Section Definition of the vector field).

Studying mobility in shorter time windows would be very interesting and it is an excellent idea for a continuation of this work. However, the type of data needed is different. It would be necessary information on users mobility at higher temporal resolution such as, for instance, the one provided by mobile phone records.

5. The authors made great efforts to demonstrate that the field fulfils Gauss theorem and has irrotational character. Please elaborate why these two findings are important? Does the irrotational character of filed only supports the definition of a potential? Can the two findings be connected to some actual phenomena of commuter flows? Could we obtain some insights in commuter flows based on the two findings?

The existence of a well-behaved field fulfilling Gauss's theorem and being irrotational is an important new insight on the deep organization of empirical recurrent mobility. These are properties that any model aimed at reproducing commuting flows must respect. Otherwise, the model does not adjust to reality. These models have been used for decades, and are still in use, to calculate trip demand in the planning of transport infrastructure. The implications of this work go, therefore, beyond theoretical considerations despite these are very relevant as well.

Gauss's and the rotational are the most basic theorems in field theory. They are the blocks upon which more involved results (metrics, theorems, etc) are built and this is why it is so important to prove that the vectors obtained from empirical data fulfilled both. Gauss's theorem means that the field is generated by a source and that the fluxes through surfaces must respect conservation laws. These constraints affect the flows and also the directions as we have shown in the previous answers regarding vector angles (new Figure 5, Fig S47 of the SI and Section Results, sub-Section Models of the manuscript). The irrotational nature of the field implies, as mentioned by the reviewer, that one can derive the field from a potential and viceversa, the field is univocally determined by the potential. The symmetries of the potential are also present in the field and, among other things, the dimensionality of the problem can be reduced: from a vector in every location to a scalar. Differences in the potential between points decide the direction and intensity of the mobility flows. We have not yet explored the deep meaning and symmetries of the potential because this would require a specific work. Out of the symmetries, usually, one can define invariant (conservative) quantities that play a central role in vector field theory. This is, however, beyond the scope of the present manuscript that intends only to set the basis for future analyses.

We must admit that these points were not clear enough in the previous version of the text, we have updated the conclusion section to cover them with more detail.

Minor comments:

1. The last sentence of page 3: please elaborate why radiation model receive more detailed input information.

We have modified the sentence to include an explanation of this point.

2. Please supplement a concrete definition of entering flux of vector filed T in the main manuscript. This will help readers better understand Fig. 4 b and d.

We have included the definition as requested in equations 5 and 6 of the Methods Section.

3. Which model did the authors use to obtain the results in Fig. 5?

It was obtained using the gravity model, we have modified the caption of the figure to leave this clear.

Reviewer #2 (Remarks to the Author):

The authors addressed successfully the points I raised in my previous report, so I recommend publication.

With best regards,

Bernat Corominas-Murtra

Reviewer #3 (Remarks to the Author):

The authors have done extensive work to improve their presentation and results. All my questions have been well addressed. I suggest the paper be accepted by Nature Communications.

In this paper, the authors propose a very interesting and new idea of employing vector field to analyze human mobility, which I believe, represents a great contribution and has great potential to open new avenues in the very active field of human mobility.

With best regards,

Pu Wang